# Comparison between constant and variable chlorine decay models with respect to initial chlorine dosage applied to urban water supply network

Ababu T. Tiruneh[1], Tesfamariam Y. Debessai [2], Gabriel C. Bwembya[2], Stanley J. Nkambule[1]

[1] University of Eswatini, Department of Environmental Health Science, P.O. Box 369, Mbabane, Eswatini.
[2] University of Eswatini, Department of Chemistry, Private Bag 4, Kwaluseni, Eswatini.

*Correspondence to*: Ababu T. Tiruneh (ababute@gmail.com)

**Abstract.** Monitoring of chlorine residual in water distribution systems is necessary not only for ensuring potable water quality but also prevent emergence of disinfection by-products due to excess chlorination. Modelling work for chlorine residual was carried out for water supply distribution network of a town using both second order and first order reaction rate models. For the development of the model, the bulk reaction decay rate was determined in the laboratory using bottle testing while the wall decay rate was determined by calibration of the water quality model using field residual chlorine concentration measurements. The model results show that there is no significant difference in the residual chlorine between the two models or the cost saving that result in terms chlorine usage for the range of initial chlorine dosages anticipated. Constant rate chlorine model is more conservative and offers additional safety in terms of chlorine residual present. Significant differences only occur at excess chlorine residual concentration within the distribution system above the intended maximum residual to be attained. Further research that relates the chlorine dose with the water quality characteristics is necessary to make a more general evaluation.

## 1. Introduction

Chlorination of water is often the last step of treatment undertaken in conventional water treatment processes in order to ensure that water is safe for consumption and reduce the risk of water borne diseases (Robescu et al., 2008; Bocelli *et al*., 2003; Fisher *et al*., 2004). Disinfection with chlorine is a widely used practice for urban water supplies worldwide and historically has been observed to prevent the instances of occurrences of waterborne diseases in cities. With the use of gas chlorine, transportation cost of chlorine reduced considerably (White, 1972).

Over time the variables affecting chlorine reaction, experimental procedures for determining chlorine dosages were established. With the risk of formation of disinfection by products (DBPs) as result of excess chlorination in which these byproducts have been linked with carcinogenic, mutagenic and toxicity property, the need for controlling the dosage of chlorine became important (Chang *et al*., 2006). In addition, other safer disinfectants such as ozone came in to use (EPA, 1974). Chlorine has a broad function beyond disinfection including control of taste and odour producing microorganisms, iron and manganese removal as well as taste and odour control (Vhutshilo *et al*., 2017). The importance of chlorination cannot be overstated particularly with the increasing pollution of water sources coming from agriculture and industry (Barakat *et al*., 2005).





In the past, a number of chlorine residual models have been proposed ranging from the simple model to the more complex ones that take account of a multitude of factors affecting the chlorine decay rate (Kohpaei and Sathasivan, 2011). The first order kinetic model of chlorine decay is given by;

$$\frac{dC}{dt} = -kC \qquad (1)$$

In Equation (1), C is the concentration of chlorine residual at time t and k is the first order reaction rate constant. The first order model is easy to solve analytically through mathematical integration of Equation (1) and has been in wide use in the past. However, it has been reported to have limitation in several applications as it discards factors such as the initial chlorine dosage and the level of reactant species present (Borcelli, *et al*., 2003; Clark, 2998; Gang *et al*., 2003; Hass and Kara, 1984). In order to overcome this limitation, a number of chlorine decay models have been proposed such as the $n^{th}$ order power decay, first order model with stable components and parallel first order decay model (Huang and Mcbean, 2007). However, all these models were pseudo first order reaction models whereby the chlorine decay rate was still dependent on the chlorine residual and time while the chlorine consuming reactant was assumed to be constant and much larger than chlorine so their concentration did not influence the decay rate. This assumption was reported to have limitations in some practical applications (Borcelli *et al*., 2003).

The two constituent decay models were an improvement over the pseudo first order chlorine decay models in which the effect of both the concentration of chlorine as well as the concentration of reactant species were taken into account in the model formulation. The earlier application of this model was the one proposed by Clark (1998) which considered a single reacting species such as THM reacting with chlorine. Clark's second order model was based on the equation:

$$C(t) = \frac{C_0(1 - R)}{1 - Re^{-ut}} \qquad (2)$$

In Equation (2), C(t) is the chlorine residual at time t, $C_0$ is the initial chlorine dose. R and u are the model parameters that are explicitly stated to be functions of the chlorine concentration and the concentration of Trihalomethanes (THM) (reactants) as well as the stoichiometric coefficients a and b stated in the chemical reaction between chlorine and Trihalomethanes. According to Clark, R and u were simply determined from the chlorine decay data by least square fit of the C(t) versus time data using the Marquardt-Levenberg method (Clark and Sivaganesan, 2002). This second order model gave results that were comparable or better results when compared with the parallel second order model and $n^{th}$ order models (Vasconcelos *et al*., 1996). They certainly gave better fit compared to the first order models. The difference with first order model varied according to the clarity of the water with respect to the total organic carbon (TOC) concentration. Clear water with low TOC concentration have slow kinetics while poor quality water with high TOC concentrations have the initial reaction rate much faster and the overall reactions thus approaching second order. The Clark model also enables estimation of the THM with time. However, this model being based on a single reactant (such as THM) reacting with chlorine, this situation may not be realistic where there are often multitude of chlorine consuming reactants that may be present in water. In



addition the model parameters u and R are valid only with respect to the initially assumed reactant concentration $C_{R0}$. If this value changes, the model parameters are expected to change as well.

The idea of two competitive reactions, namely, fast and slow reactions came into focus with the need for models to
accommodate these two different reaction phases at the same time. Clark and Sivaganesan (2002) attempted to extend the second order model developed by Clark (1998) by dividing the chlorine consumption in to two components namely chlorine consumed by fast reactants and that consumed by slow reactants. The model just adds up Equation (2) twice for the fast and slow reacting species. As a result five model parameters were to be determined in an iteration process involving three consecutive steps. In addition, the model parameters were further
related to TOC, UV absorption, pH, alkalinity, bromide concentration and temperature. However, this model, being based on artificially splitting and apportioning the chlorine residual into two for the fast and slow reacting species, deemed unrealistic as what was present was a single concentration of chlorine residual available for both the fast and slow reacting species.

Philip *et al* (2009) considered the more realistic case of the presence of several reactants at the same time and defined the concentration weighted reaction rate $k_t$. However, in trying to express the variation of $k_t$ with time, the rate of reaction of individual reactants as well as their molar concentration still appear in the equation although the second order nature of the reaction was apparent. Instead they proposed the following empirical equivalent equation for their model;

$$\frac{dk_t}{dt} = aC_t\left(k_t k_{min} - k_t^2\right) \qquad (3)$$


There are four model parameters in Equation (3) namely the value of empirical constant a , the rate constant of the slowest reactant $k_{min}$, the initial overall concentration weighted reaction rate $k_0$ and the initial concentration of reactants $X_0$.   The model parameters were determined from experimental data by numerical solution of the differential equation given by Equation (3) using Euler method and choosing the optimum values of the model
parameters through genetic algorithm.  Tiruneh *et al*. (2019)  modified the model developed by Phillip *et al* (2009) and provided a mathematical model for the reaction rate variation with time as a function of the chlorine residual, the rate constant itself and the ratio of arithmetic to harmonic mean of the concentration of reactants as given in Equation (4).  The three parameters to be determined are the initial concentration of reactants $X_0$, the initial rate of reaction $k_0$ and the ratio of arithmetic to harmonic mean of the concentration of reactants $R = \frac{\bar{X}_A}{\bar{X}_H}$

$$\frac{dk_t}{dt} = -C_t k_t^2 \left(\frac{\bar{X}_A}{\bar{X}_H} - 1\right) \qquad (4)$$


Tiruneh et *al*. (2019) also further developed the second order model for the variation of the rate constant with the initial chlorine dosage according to Equation (5). The model parameters β and $K_0$ in Equation (5) were developed experimentally by running bulk decay of chlorine test at different initial chlorine doses and using linear regression



analysis after equation (5) was linearized with respect to the rate constant k and the initial chlorine dose $C_0$. The
result of this model is also applied in this paper for modeling the chlorine residual in the distribution system using
variable chlorine decay modeling.

$$K \ = \ \frac{\beta}{1 \ + \ \beta K_0 C_0} \qquad (5)$$

Fisher *et al*.( 2011[a]) developed a two reactant (2R) second order model as a simple general model in which the fast
and slow reacting reactions are expressed in Equation (6):


$$\frac{dC}{dt} \ = \ \frac{dC_f}{dt} + \frac{dC_s}{dt} \ = \ -k_f C C_f - k_s C C_s \qquad (6)$$

Where $C_f$ and $C_s$ are the concentrations of fast and slow reacting species and $k_f$ and $k_s$ are their respective decay rate
coefficients. This model requires estimation of these four parameters, namely, $C_f$, $C_s$, $k_f$ and $k_s$. Their values are
determined from laboratory decay test data. This model has recently been incorporated in the EPANET software as a
multi-species extension (EPANET MSX) for modeling chlorine residuals as a second order, two reactant reaction
(Monteiro *et al*., 2014). Kohpaei, A.J. and Sathasivan (2011) presented analytical solution of Equation (6) for the
parallel 2R model in which the solution becomes a sum of two serial reactions modeled according to Equation (2)
developed by Clark (1998) where by the effect of the fast and slow reactant concentrations are discounted by a set of
valid assumptions. For example for the fast reaction the effect of the slow reactant is negligible as it remains
constant during the initial period of fast reaction.

### 1.1. Water quality model for chlorine residuals

A number of water quality model programs for modelling residual chlorine are available some of which are
proprietary and others are free and can be downloaded from the source site. The EPANET is one such freely
available program developed by USEPA and that was written with the C programming language. It has been used
widely and is reported for its reliability (Rossman, 2000; HDR Engineering, 2001). The hydraulic network analysis
in the EPANET program is based on gradient method while the water quality model for reactive chemicals is based
on mass balance equation that take account of bulk transport within the pipes, mixing between pipes, storage tanks
and chlorine decay reaction that occurs within the bulk water and the pipe walls. The EPANET water quality model
requires specification of the initial chlorine dosage, the bulk chlorine decay rate and the wall chlorine decay rate.
(Mohamed et *al*., 2018). The wall chlorine decay rate is modelled based on molecular mass transfer for chlorine,
mass of chlorine present in the bulk water and the wall decay rate of chlorine at the pipe walls. The bulk decay rate
is determined through laboratory bottle tests and the wall decay rates are established commonly through calibration
of the hydraulic model with field data for chlorine residual (Haider , *et al*., 2015).



The EPANET program has limitation as it uses monolithic values for bulk and wall decay rate constants of chlorine. This may not be true owing to variation due to pipe material, pipe age, pipe diameter, biofilm thickness and type and other factors that may vary from length to length within the distribution system (Foong et al., 2004). Besides, there are also questions raised regarding the validity of simple models that are based either on first or second order rate variation (Clark, 1998; Kastl *et al*., 1999). Alternative models have been developed such as the two reactant model

by Fisher *et al*. (2011[b]) involving fast and slow reactions. An extension developed within EPANET allows users to define reactions that are suitable for wall and bulk decay (Sahng, 2008).

    The objective of this research was to examine the influence of variable chlorine decay modelling such as the one provided by Equation (5) on the chlorine residual in distribution system compared to constant rate chlorine

modelling that is routinely carried out using the EPANET program. This was the first step for the Matsapa distribution network where this research is being carried out. As a further research in future, the influence of the water quality characteristics on the chlorine decay model to be used and the resulting chlorine residual as well as residual reactants will be studied.

## 2.       Materials and Methods

The water quality modeling work was carried out for the Matsapha area water distribution of network that is shown in Figure 1. The network consists of treated water that is pumped from the water treatment plant to two storage tanks and a distribution network consisting of 36 pipes and 28 nodes. The water flows to the consumers from the storage tanks. The raw water source is river water and undergoes treatment consisting of screening, chemical addition, coagulation, flocculation, settlement, filtration and finally disinfection. The average daily flow of water from the

treatment plant to the distribution system is about 5 million liters per day.

    The network map was imported to the EPANET program and the information required for hydraulic and water quality modeling was entered in the EPANET platform. The bulk decay rate constant of chlorine was determined for several initial chlorine concentrations using laboratory bottle tests. As will be shown in the results section, a

second order decay rate model with respect to the initial chlorine dosage adequately models the decay rate. During the bulk decay rate test, iodometric titration was used for the determination of chlorine residual at different times and for a given initial chlorine dose added to the water samples taken from the treated water after filtration but before chlorination. The standard operating procedure stated in the Standard Methods for the Examination of Water and Wastewater (APHA, 1999) was used for the determination of chlorine using the iodometric titration. The wall

decay rate constant was determined using field measurement of chlorine residuals taken at four points within the distribution system. The measurements were made on site using mobile chlorine measurement instrument. The wall decay rate constant was determined by trial and error procedure in which different values of the wall decay rate constants were entered in the EPANET and the water quality model run for each rate constant. The wall decay rate



constant that gave the minimum total least square error between the model residual chlorine and the field determined

values of residual chlorine was taken for the modeling exercise.

The EPANET calibration was carried out when the initial chlorine residual in the treated water entering the distribution system was 1.0 mg/L. This means that at this rate of initial chlorine dosage both the constant rate and variable rate chlorine models have the same rate of decay and would produce the same result in the EPANET

modeling. Then after for comparison of the results the EPANET program was run at different initial chlorine dosage from the one used for calibration and the result of the model output in terms of chlorine residual were compared with the model output of the constant rate model. It should be noted that this analysis takes only the initial chlorine dosage as a factor in the variable rate modeling assuming that other factors such as reactant species, temperature, pH are the same. The modeling of such factors is not treated in this paper and needs to be considered differently as

provided in the introduction of this paper such as by employing the two reactant (2R) model.


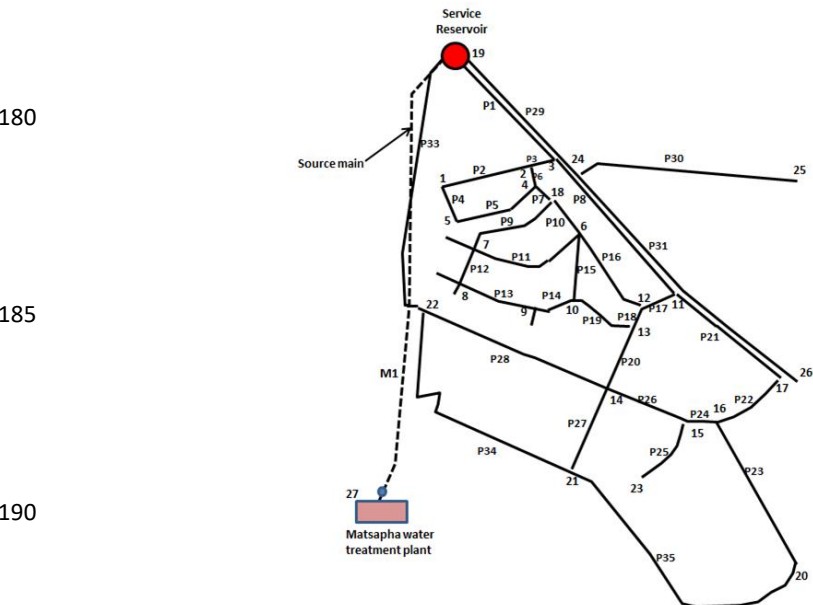

**Figure 1: Pipe network layout**

**3.    Results  and discussion**

**3.1.    Initial chlorine dose modeling**

The result of the second order modeling of chlorine decay rate with respect to the initial chlorine dosage based on experimental result of the bulk decay rate for different initial chlorine concentrations is shown in Figure 2. It is clear



from the figure that the reaction rate decreases with the initial chlorine. Moreover, the rate of decrease is faster at low initial chlorine dosages. This means that at high initial chlorine dosage the reaction tends to approach first order reaction compared to low chlorine dosages. The second order model parameters of Equation (5) were determined by linear regression. The regression fit gave a coefficient of determination $R^2 = 0.99$. Equation (7) gives the variation of the bulk chlorine decay rate constant $k_d$ with the initial chlorine concentration $C_0$.

$$k_d = \frac{0.580}{1 + 0.843\, C_0} \qquad (7)$$

### 3.2.   Hydraulic and water quality modeling using EPANET 2.0

The pipe network diagram for the Matsapha network shown in Figure 1 was imported to the EPANET and required data such as the nodal water demand, pipe length, diameter, roughness coefficient, ground elevation, etc., were entered as required by the EPANET. For the EPANET analysis since the two storage tanks are interconnected, they were converted to an equivalent single storage tank having the same volume and water height.  In order to carry out extended period simulation of the flow in the pipe network, the flow records obtained from the Matsapha network

varying over a period of 24 hours were used. The daily variation of water demand was divided into six time periods each having duration of four hours. The peak factor to be used was worked out for each of the six periods and entered into the EPANET program..  The value of the bulk chlorine decay rate constant to be used was entered by calculating this constant against the initial chlorine concentration present and used in the model.

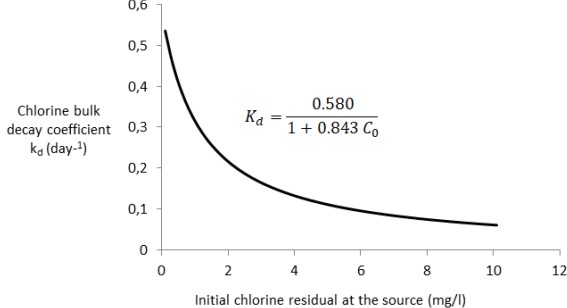


**Figure 2:  Second order chlorine bulk decay modeling curve used in the EPANET program**

### 3.3.   Determination of the wall chlorine decay rate constant

The pipe wall decay coefficient was determined using trial and error procedure by assuming different wall decay

rate constant values and running the EPANET water quality model and determining the chlorine residual at different points in the distribution system. The first order reaction rate has been used for modeling the wall decay of chlorine. For calibration purposes, field chlorine residual measurements at four different sampling points within the

distribution system were used to compare the field measurement result with that of the model output. The sum of squares of error between the model output and the field chlorine residual measurements was plotted against the assumed wall decay rate constant used in the modeling. Figure 3 shows this plot showing that minimum total square error occurs for wall decay rate constant of 0.05 per day. This value of wall decay rate constant has been used for subsequent modeling.

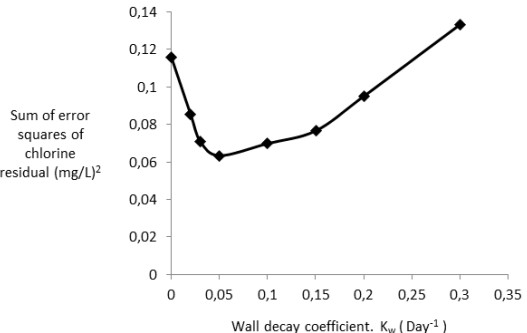

**Figure 3: Plot of sum of error squares of chlorine residual plotted against wall decay coefficient used in the EPANET model using four points selected from the network.**

In order to identify sampling points for analysis, extended period simulation was run for a period of 288 hours. Figure 4 shows the plot of water age at the four sampling points selected afterwards. Sampling point at Spintex (Node 20) has the longest water age as it is located furthest from the service reservoir in the network. The other three points included SEC (Node 21) Old Airport (Node 24) and Tubungu (Node 25).

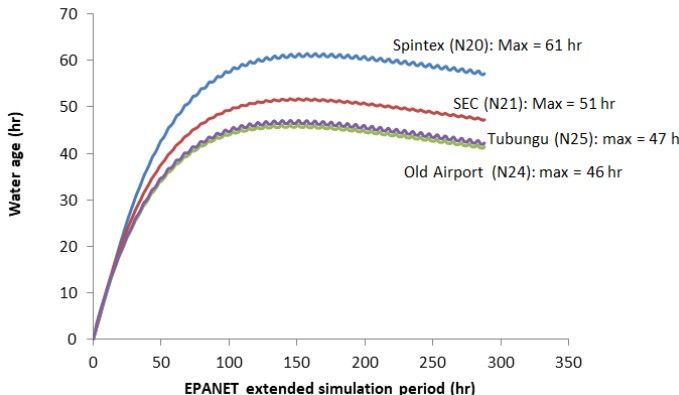

**Figure 4: EPANET extended period simulation of water age among the four sampling points within Matsapha network**

**3.1.    Comparison of chlorine residual model results between constant and variable chlorine models**

Figure 5 and Figure 6 show the results of the model simulation for chlorine residual for both the variable and constant rate models. It can be seen that overall the constant rate model tends to underestimate the chlorine residual requiring higher initial chlorine dosage at the source for a given desired chlorine residual at demand points within the distribution. The constant rate model is therefore a conservative design. On the other hand, the difference
between constant and variable chlorine model results is not very significant. Figure 5 and 6 show that for all the four sampling points within the range of the desirable residual concentration at demand nodes, both constant and variable chlorine models give comparable values for the desired initial chlorine dosage required at the source. The difference only becomes greater when the residual concentration is high which is not desirable as excess chlorine dosage leads to formation of disinfection by products beside posing taste and odour problem to consumers.


Figure 7 shows the percentage error that occurs in modeling using the constant rate model against the variable rate model as is obtained from the extended period analysis against the minimum chlorine residual. The minimum residual is chosen because it occurs at longer water age and magnifies the difference in chlorine consumption between the constant and variable chlorine models.  While for the ranges of initial chlorine studied the difference is
within 15%, it can be seen that within the practical maximum  initial chlorine dosage of less than 3 mg/L, the percentage error for all the four sampling points are less than 10%. The error varies with the distance from the network. Node 20 (Spintex) has the highest error as it is located longest distance followed by SEC (Node 21), Tubungu (N25) and Old airport (N24) in that order.

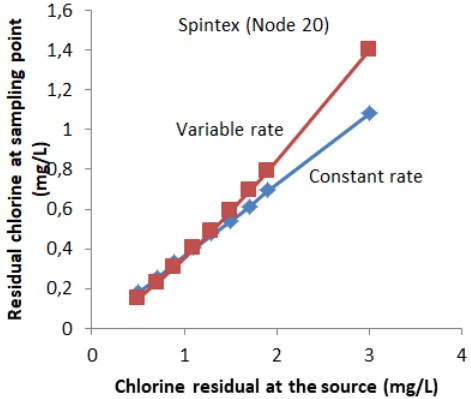
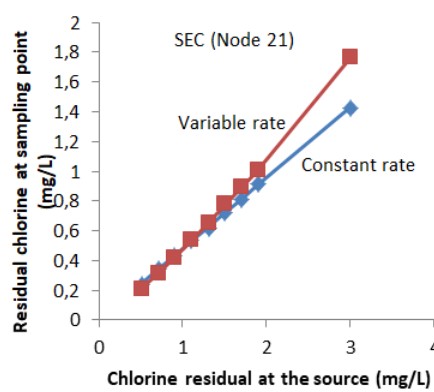

**Figure 5: Residual chlorine at Spintex (Node 20) and SEC (N21) showing EPANET output using two alternate models**

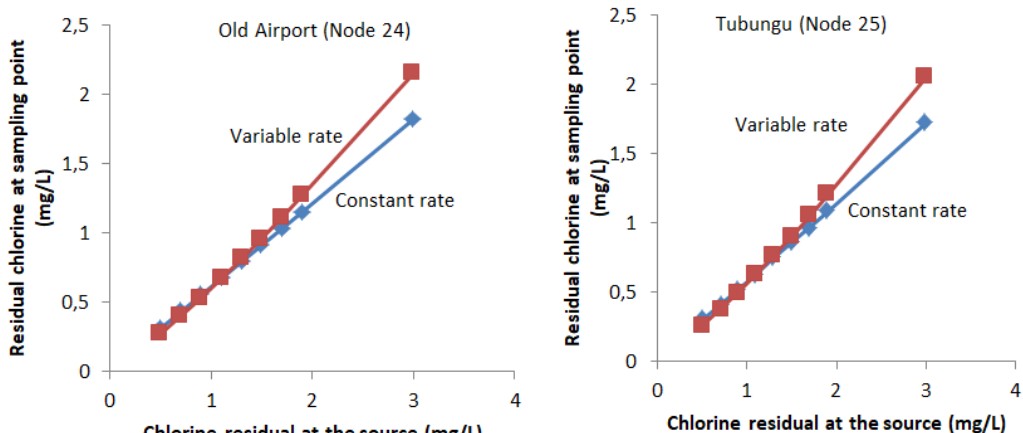

**Figure 6: Residual chlorine at Old Airport (Node 24) and Tubungu (Node 25) showing EPANET output using two alternate models**


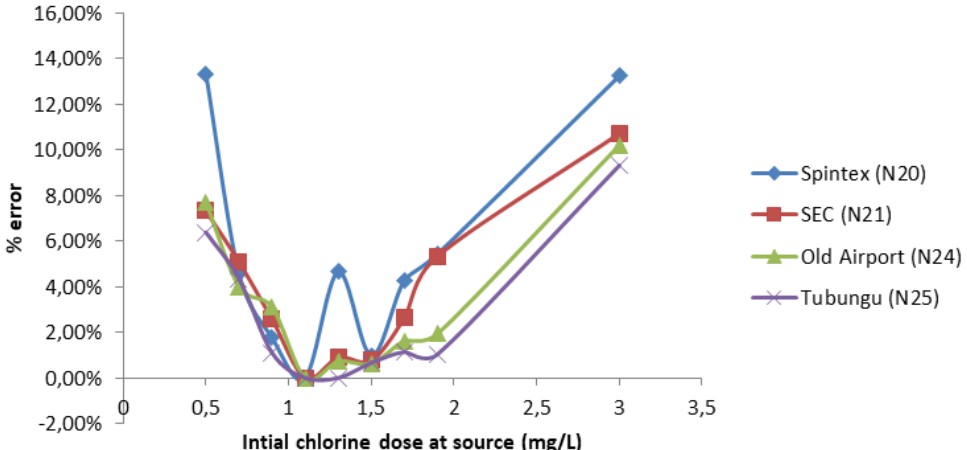

**Figure 7:  Percentage error of constant rate model against variable rate at minimum residual**

The experimental results of the variation of the reaction rate with initial chlorine suggest that the reaction rate varies
significantly with initial chlorine dosage used. This variation is observed to be higher at low initial chlorine dosages.
However, despite such variation in reaction rate as suggested by the second order model result, the difference
between the constant and variable (second order) model with respect to the initial chlorine dose is not big as applied
to the Matsapha network. The high quality of the treated water may be a factor as it exerts low chlorine demand. In



general, the comparison of first and second order model with respect of the initial chlorine dosage can be variable
depending on the quality of the water. While the second order model produces comparatively better data fit, the
difference with the first order model may not be big (Al Heboos and Licskó, 2017). On the other hand poor quality
water with high level of dissolved organic carbon may influence the rate of reaction in such a way that the results of
the second order model results are significantly different from that of the first order. This was cited earlier in this
paper based on the work of Clark (1998). The results obtained in this paper may have narrow range of application
since when the water quality characteristics change, the results of the comparison between constant and variable rate
models may also change. It is, therefore, necessary as a further research to examine the combined effect of the
reactant characteristics and the initial chlorine dosage on the rate of reaction. This can be carried out by considering
the raw water characteristics over the different seasons or considering dilution experiments in which the raw water is
diluted with reactant free water and the chlorine decay experiments are carried out for this low reactant water.

**4.     Conclusion**

Chlorine residual modeling in water supply systems is a useful exercise for management of initial chlorine dosage at
the source as well as chlorine residual within the distribution system. Currently the chlorine dosage within the
Matsapha network is being monitored for chlorine residual through periodic checking of the residuals at different
points and subsequent adjustment of dosages at the source. The hydraulic model presented offers a better approach
for routine monitoring purposes. According to the results of the model presented, there is no significant difference
between constant and variable chlorine modeling within the range of initial chlorine dosage used in the system as
well as the range of residual chlorine concentrations required to be achieved at the user points. The difference
become significant for high concentration of chlorine residuals typically exceeding 1 mg/L which is not desirable
from the point of view of production of disinfection by products that can be health hazards of taste and odour
problem due to excess chlorine. The constant rate modeling is easier to establish. The only required parameter to be
determined is the average bulk chlorine decay rate constant for the range of initial chlorine dosage anticipated.
Variable chlorine modeling requires repeated experiment at different concentration in order to establish the
relationship between the bulk chlorine decay rate constant and the initial chlorine dosage used. It is, therefore, more
involved. In addition, constant rate models are conservative because the models demand higher initial chlorine
dosage at the source for given minimum chlorine residual to be satisfied at the demand point. They, therefore,
provide additional safety factor against possible deterioration in chlorine residual particularly at far ends of the
distribution system. However, because the condition of the treated water may change from season to season there is
a need to study together the water quality characteristics together with the chlorine dosage in order to evaluate the
difference between these two models.

**Acknowledgment**



The authors would like to thank the Eswatini Water Services Corporation (EWSC) for giving permission to carryout field testing of chlorine residuals within the Matsapha town water distribution network and supplying hydraulic network data.

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
