# Peer review of "Comparison between constant and variable chlorine decay models with respect to initial chlorine dosage applied to urban water supply network"

_Drinking Water Engineering and Science, 2020_

## Referee Comment (RC1) · Anonymous Referee #1 · 10 Aug 2020

This paper compares chlorine residual modelling in a water distribution system using a first and second order reaction rate models. It is well written and I enjoyed reading it.

For publication in this (or any other) scientific journal, a manuscript needs to "represent a substantial contribution to scientific progress within the scope of this journal (substantial new concepts, ideas, methods, or data)". Unfortunately I can't see any such contribution in the paper. The two chlorine decay models are well known, the laboratory tests used to calibrate these models are routine and the application of the model to the Matsapa distribution network is done without reference to measured chlorine residuals in the field.

[Figure]

I would like to encourage the authors to do further work and resubmit the paper when they have added a novel contribution. Given that this supply network is in a rural area serving a particular demographic, and that it seems the local supply authority doesn't have the resources for detailed monitoring and operations, expanding the paper to describe the challenges faced with this research and how they were be overcome will add novelty to the paper and make it useful to people working in similar areas. It is essential to take residual chlorine measurements in the field and compare the simulations models to actual data, rather than simply comparing the results of two simulations models.

---

## Author Comment (AC1) · 11 Aug 2020

We thank thr reviewer for for forwarding the evaluation. With due respect to the comment forwarded this is to make further clarification (not necessarily a defence against the comment) that the authors suggested a second order model for the variation of the decay rate constant with intial chlorine dose. The lireature availble mainly indicated emperical inverse relationship between the decay rate constant and the intial chlorine dose of the form $K = 1/(a+bC_0)$ where $C_0$ is the intial chlorine and $K$ is the first order decay rate constant. In many cases a graphical description or specification of a numerical range is given to narrate the relationship between intial chlorine and the first order

rate constant. We believe that our model started from the theooretical foundation of the dependence of the first order decay rate constant with intial chlorine modelled as a second order reaction of the form dK/dt = mK^2 in which the empirical inverse relation that we mentioned will be a consequence of this general model. It might be helpful to look at the theoretical background of our earlier paper

https://www.hindawi.com/journals/mse/2019/5863905/

that a gives a comprehensive theoretical justidfication of our proposed model of the second order nature of the variation of the first order decay rate constant with the intial chlorine dose. We also would like to state that present paralel two reactant second order models work with given intial chlorine dose along with water quality paremeters such as chlorine consuming reactantns. in other words, the fast and slow reaction rates Kf and Ks deribed from the model change whenever the initial chlorine dose changes. Leading authors recently suggested a way of handling this variation in this parallel two reactant model that takes accunt of the effect of intial chlorine dose. In this line it may be helpful to look at the paper by one of the leading researchers (Ian Fischer) who cited the need for research in this direction with the following link:

https://ascelibrary.org/doi/abs/10.1061/%28ASCE%29WR.1943-5452.0001101

We believe that our formal second order model over and above the empirical inverse relation we mentioned can be integrated in such models. We also compared the practical range within which the constant and variable rates modelling differ based on the model we developed and applying it to a real case of a water distribution system and provided the perecentage range within which the two models differ and the range of initial chlorine dose for which the percentage difference between the two models is/not significant. Several other researchers also indicated the need for formal modelling approach to the effect of intial chlorine dose and water quality on the decay rate constants suggesting a transition from the now mostly used empirical, regression based relationship.

---

## Referee Comment (RC2) · Anonymous Referee #2 · 14 Aug 2020

Two models were compared in a full scale network to estimate chlorine decay. General comments: - It is not clear what the novelty of this study is. - The manuscript reads as a report of an engineering project - The introduction is too long and contains too much text book knowledge - The results should be discussed with literature and new insights should be highlighted - The study should be extended e.g. as the authors suggested: "It is, therefore, necessary as a further research to examine the combined effect of the reactant characteristics and the initial chlorine dosage on the rate of reaction. This can be carried out by considering the raw water characteristics over the different seasons or considering dilution experiments in which the raw water is diluted with reactant free water and the chlorine decay experiments are carried out for this low reactant water". -

This should be hand in hand with a clear scientific question to be answered.

---

## Author Comment (AC2) · 14 Aug 2020

We are thankful to anonymous referee #2 for forwarding the comments posted in the interactive section and here is our response to those comments.

Here is our reponse to referree #2 comments:

1. On the claim by the refreree of our research looking lie " a research report" we dispute the claim and , as opposed to engineering reports that are commonly "procedural" we feel that our work is a research report that has engineering element in it and may probably intesect with an engineering report that has research element in it. We

have followed a research protocol of developing a new model to describe the influence of intital chlorine dose on the chlorine decay rate kinetics and experimentally verified the results. we used this experimentally verified model of variable kinetics to test and compare against constant rate models using available water quality modelling program EPANET. we high lighted the regimes where significant differences are. We believe we addressed relevant resaerch prblem as such.

2. On the introduction being too long, we would like to refer the referee to the introduction section which only briefly highlights the relevant research achievements. Of course we only gave a brief background as is the case with almost all similiar publications related to chlorine decay. secondly we cited the single reactant pionering model of clark and highlighted where the intial chlorine dose affects the models. Thirdly we mentioned the fischer model that improves to a two reactant model. Finally we high ligt the alternative variable decay mode of phillip (2009) and added our own contribution (Tiruneh et al, 2019). That is the introduction as far as it goes. All that were mentioned are brief and specifically only in relation to the research developments. There is no refernce to any text book or report. It is a brief citation of research relevant works as they influnce the theme of our particular research. It would be helpful to look at dozens of similar research papers dealing with chlorine decay and the approach and style of treatment of the introduction section in our paper is not any different.

3. On the question of novelty, we would like to refer the referee to our reply to referee #1 to avoid repetition in which we highlighted our contribution in this research. Since novelty is important in research paper evaluation yet is such a "loaded' term we suggest that it would be helpful to contextual such comments with what has been done in the past and why this paper is not any different so as to avoid the risk of unfairly stereotyping what is otherwise a fairly relevant reserach output.

4. On discussion and new insights, we have included aqs far as we can and as far as the results of our reaerach enabled us in the discussion and conclusion sections and have contextuaised our research result with similar research done in the past and

highlighted the need for further additional research.